# Data-driven approach for the delineation of the irritative zone in epilepsy in MEG

**Valerii Chirkov**[1], **Anna Kryuchkova**[2], **Alexandra Koptelova**[2], **Tatiana Stroganova**[2], **Alexandra Kuznetsova**[3], **Daria Kleeva**[3], **Alexei Ossadtchi**[3], **Tommaso Fedele**[3]*

1 Berlin School of Mind and Brain, Humboldt University, Berlin, Germany, 2 Center for Neurocognitive Research, MEG Center, MSUPE, Moscow, Russian Federation, 3 Institute of Cognitive Neuroscience, National Research University Higher School of Economics, Moscow, Russian Federation

* fedele.tm@gmail.com

**Data Availability Statement:** We have reserved a DOI on the Zenodo platform 10.5281/zenodo.7114578.

## Abstract

The reliable identification of the irritative zone (IZ) is a prerequisite for the correct clinical evaluation of medically refractory patients affected by epilepsy. Given the complexity of MEG data, visual analysis of epileptiform neurophysiological activity is highly time consuming and might leave clinically relevant information undetected. We recorded and analyzed the interictal activity from seven patients affected by epilepsy (Vectorview Neuromag), who successfully underwent epilepsy surgery (Engel >= II). We visually marked and localized characteristic epileptiform activity (VIS). We implemented a two-stage pipeline for the detection of interictal spikes and the delineation of the IZ. First, we detected candidate events from peaky ICA components, and then clustered events around spatio-temporal patterns identified by convolutional sparse coding. We used the average of clustered events to create IZ maps computed at the amplitude peak (PEAK), and at the 50% of the peak ascending slope (SLOPE). We validated our approach by computing the distance of the estimated IZ (VIS, SLOPE and PEAK) from the border of the surgically resected area (RA). We identified 25 spatiotemporal patterns mimicking the underlying interictal activity (3.6 clusters/patient). Each cluster was populated on average by 22.1 [15.0–31.0] spikes. The predicted IZ maps had an average distance from the resection margin of 8.4 ± 9.3 mm for visual analysis, 12.0 ± 16.5 mm for SLOPE and 22.7 ±. 16.4 mm for PEAK. The consideration of the source spread at the ascending slope provided an IZ closer to RA and resembled the analysis of an expert observer. We validated here the performance of a data-driven approach for the automated detection of interictal spikes and delineation of the IZ. This computational framework provides the basis for reproducible and bias-free analysis of MEG recordings in epilepsy.

## 1 Introduction

Magnetoencephalography (MEG) has been proven as a useful clinical tool for the improvement of surgery outcome in epilepsy [1,2]. Guidelines are provided for the use of MEG in clinical settings [3–5] and its integration in the presurgical work-up [6]. The impact of MEG in epilepsy investigation has been retrospectively validated on intracranial EEG recordings on large patient cohorts, with resection of MEG foci associated with good outcome [7,8]. The

**Funding:** V.K., A.G., and T.F. were supported by Russian Foundation for Basic Research, N°20-015-00176 A (https://kias.rfbr.ru/). This work/article is an output of a research project implemented as part of the Basic Research Program at the National Research University Higher School of Economics (HSE University) and was carried out using HSE Automated system of non-invasive brain stimulation with the possibility of synchronous registration of brain activity and registration of eye movements. The funders had no role in study design, data collection and analysis, decision to publish, or preparation of the manuscript.

**Competing interests:** The authors have declared that no competing interests exist.

consideration of MEG findings in patient treatment prospectively improved clinical management optimizing intracortical EEG implantation and surgical plan [9,10]. This evidence is built on the localization of interictal epileptiform discharges (IED), in the form of spikes and sharp waves. While interictal activity contains valuable information to support successful non-invasive clinical assessment [11], the identification of epileptiform events is committed to the visual inspection of expert reviewers. MEG recordings are characterized by multivariate information embedded in some few hundreds channels, where possibly repeating patterns with variable signal-to-noise ratio (SNR) arise from the background activity. In this context, visual analysis represents a time consuming procedure prone to human bias.

Several automated approaches for IED detection have been proposed for invasive and non-invasive EEG, based on a-priori assumptions on the morphological, spectral and statistical properties of epileptic spikes [12]. Recent approaches based on machine learning showed high sensitivity of linear classifiers [13,14] and deep-learning [15] for the detection of visually identified events. These strategies have been applied retrospectively, based on specific properties of the spike signal or following the training of a classifier on extensive available datasets. This poses some limitations on their applicability to prospective cases.

Ideally, MEG analysis requires data-driven adaptive strategies, scalable to the peculiarity of the individual case. An interesting approach is this direction is represented by dictionary learning [16]. The target of this approach is the extraction of prototypical waveforms, to which we refer here as 'atoms', that are emblematic representations of the signal under study. The mathematical framework to apply dictionary learning of atoms to multivariate neural time series is provided by convolutional sparse coding (CSC, [17]). While CSC has been successfully applied to image processing and neurophysiological signals [18], an implementation particularly suitable for MEG has been recently proposed [19,20].

Here we tested and validated a data-driven strategy, which aims to identify and cluster clinically relevant events, with minimal assumptions imposed by the user and with the option of visual validation of extracted clusters. Our approach is organized in two main stages: first, the selection of candidate IED by peak detection and dipole fitting [21]; second, the validation of candidate IEDs undergoes spatiotemporal clustering by CSC [19]. We applied this pipeline to MEG data of epilepsy patients who underwent successful surgical resection and quantified the clinical relevance of the automatically identified irritative zone with respect to the outcome of the IED visual analysis and the resected area.

## 2 Methods

The analysis workflow is shown in Fig 1.

### 2.1 Patients

Patients were recorded between 2012–2018 at the MEG Center of Moscow State University of Psychology and Education (MSUPE), Moscow, Russian Federation. We have selected 7 patients (N male = 2, average age 15.8) meeting the following inclusion criteria: 1) The patient underwent focal epilepsy surgery after the MEG recording; 2) Follow-up > 1 year, with known outcome according to the Engel scale; 3) Availability of an individual pre-surgical MR; 4) Description of the resected area, or alternatively, availability of the post-surgical MR; 5) Availability of visually marked epileptic spikes. Detailed clinical characterization for the entire dataset is provided in Table 1. The selection of the putative seizure onset zone was based on the combined analysis of the interictal and pre-ictal events in these non-lesional or complex-lesional patients. Then they underwent pre-surgical or intra-operative ECoG based on the localization given by presurgical source localization of the MEG events. For a more detailed description see [22].

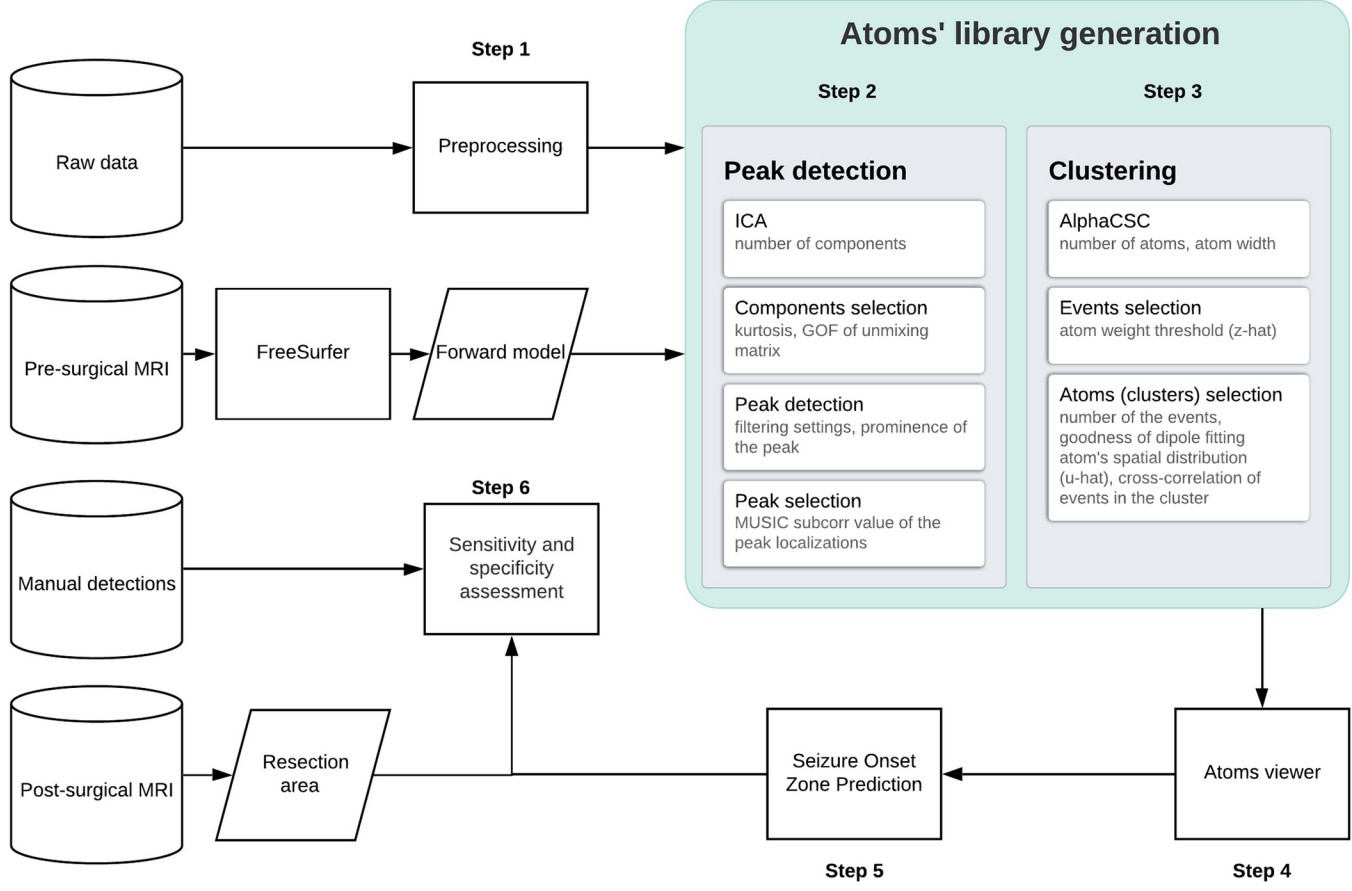

**Fig 1. Workflow overview.**

Patients signed the written consent to participate in the study. This study was approved by the ethical committees of Moscow State University of Psychology & Education (MSUPE), Moscow, Russia.

## 2.2 MEG data acquisition

Patients were recorded during sleep accommodated in supine position with the head fitting a 306-channel MEG system (102 magnetometers and 204 gradiometers, Elekta Neuromag Oy, Helsinki, Finland) at the MEG Center of MSUPE. Data was acquired at a sampling rate of 1000 Hz. Additionally, ECG, EOG and EMG from the right and the left masseter muscles were recorded. Hands and legs movement were tracked with four accelerometers fixed at the index fingers of both hands and the second toe of both feet. Head movements were monitored by four head position indicator (HPI) coils, whose position was constantly tracked during the recordings. The 3 anatomical fiducial points (nasion, left and right preauricular points) also were digitized using FASTRAK system (Polhemus Inc., Colchester, VT, USA) for the MRI-MEG co-registration.

## 2.3 MEG data preprocessing

MEG data preprocessing was performed according to the standard practice for clinical MEG research [23,24]. The data were processed by MaxFilter (Elekta Neuromag Software) using temporal signal space separation (tSSS, [25]). For each patient, out of the 1–2 hours available

**Table 1. Patients' description.**

| Patient | Etiology (biopsy) | Age/ gender | SOZ (MEG) | N visual spikes | MRI findings | Additional assessment | Surgery | Resection area | Outcome (Engel) | Follow up (months) |
|---|---|---|---|---|---|---|---|---|---|---|
| 1 | FCD IIIc | 16/m | Right T Bas | 62 | 1)FCD I in Right medial occipital area 2)FCD in Right parahippocampal gyrus 3)Thinning of Right insula | 1)PET: hypometabolism in Right TL, Right PL, Right insula 2)Stereo-EEG before MEG: presumably Right TL. | tailored resection | Right T Bas | IA | 12 |
| 2 | FCD I, FCD II | 8/f | Left F Bas+Lat | 55 | MR-negative | 1)Ictal SPECT: Left Bas, 2) PET: Left Bas 3)Intraoperative ECoG | tailored resection | Left FL Bas+Lat | IA | 48 |
| 3 | TSC | 16/f | Right T Bas+Lat | 73 | Multiple tubers | 1)Ictal SPECT: hypometabolim in Right TL 2)Intraoperative ECoG | tuber removal | Right TL | IA | 46 |
| 4 | Ganglioglioma WHO grade I | 6.5/f | F Med | 79 | glioma in Right F area | — | tumorectomy | poster FL | IA | 54 |
| 5 | HS, FCD Ib, Right TL | 28/f | Right T | 106 | Right HS, right anterio-latero-Bas dysplasia | Intraoperative ECoG | anterio-medial resection | Right TL | IB | 78 |
| 6 | TSC | 4.9/m | Left T | 89 | Multiple tubers | Presurgical long-term invasive EEG monitoring | anterior lobectomy | Left TL | IIA | 22 |
| 7 | FCD Ic | 20/f | Left T | 99 | Right Hippocampal Gyrus | Intraoperative ECoG | anterio-medial resection | Left TL | IIB | 53 |

Bas = basal, F = female, F = Frontal, FCD = focal cortical dysplasia, HS = hippocampal sclerosis, HG = hippocampal gliosis, PET = positron emission tomography, SPECT = Single-photon emission computed tomography, ECoG = electrocorticography, L = Lobe, lat = lateral, m = male, Med = Medial, T = Temporal, P = parietal, TSC = tuberous sclerosis, WHO = World Health Organization.

recording, we selected 20 minutes with the highest number of visually marked spikes during sleep. MEG data were visually inspected and noisy segments were excluded for the automated spike detection pipeline ($<$ 2 minutes out of 20-minutes recording). Eye blinks and heart beats were projected out by ICA decomposition.

### 2.4 Anatomical data processing

**2.4.1 Head model.** Pre-surgical T1 MRI images (voxel size is 1 mm) with FreeSurfer v6.0.0 (https://surfer.nmr.mgh.harvard.edu). The MRI and MEG spaces were aligned in the MNE-Python co-registration GUI [26]. The head model was computed using a single layer (inner skull) Boundary Element Method (BEM) model with around 10200 vertices covering the cortical surface. The forward model was generated assigning freely oriented sources to all vertices.

**2.4.2 Resected area.** The resected area (RA) was delineated co-registering pre-surgical and post-surgical T1 MRI images in MNI space by Brainstorm [27]. The RA was manually delineated on the post-surgical MRI image. We created a binary mask with value equal to 1 for all sources inside the resection volume or within 3 mm from the resection border.

### 2.5 Visual marking

Epileptiform events were identified according to the glossary of terms most commonly used by clinical electroencephalographers and recommendations The American Clinical MEG

Society (ACMEGS) [4,28] (Kane et al. 2017; Bagić et al. 2011). The epileptiform discharges were reconstructed at the peak using the multi-dipole modeling procedure (ECD, Equivalent current dipole) implemented into the Elekta Neuromag software, using a spherical head model. Three expert reviewers agreed on the marked events (T.S, A.K., A.Kr.). It is important to stress that not all spikes were visually marked, but only the amount necessary to compile a patient report.

## 2.6 Automated spike detection and clustering

We present here a fully automated data-driven pipeline for spike detection and clustering. Our pipeline includes two main stages: 1) peak detection in the feature space of ICA-decomposed MEG sensor data [21] and 2) spatio-temporal clustering of the sensor-space MEG data epochs around the detected peaks based on convolutional sparse coding [19].

The pipeline was run separately for gradiometers and magnetometers, generating for each sensor type a library of clusters, with each cluster associated to a list of events. The libraries were created through an iterative procedure including stages 1 and 2 but sampling from different subsets of events. Once clusters were formed, the averages of the events within each cluster were used to map the irritative zone. In the following we provide implementation details.

**2.6.1 Stage I.** *ICA decomposition*. To reduce computing load, the data were resampled to 200 Hz. MEG data bandpassed from 2 to 90 Hz were decomposed by fastICA. The number of components was restricted to 20 for all patients and sensor types.

*Components selection*. Among the first 10 ICA components ranked by the explained variance, we selected those presenting a highly "peaky" temporal pattern, reflected by high Kurtosis values (1 to 10, see also S1 Fig), and mimicking a dipolar spatial pattern, reflected by the goodness of fit (GOF) of spatial pattern ECD (0 to 100%). For magnetometers the GOF threshold was equal to 80% and for the gradiometers to 60%. If GOF was greater than 95%, the ICA component was included irrespective of the Kurtosis value. Dipole fitting was performed by MUSIC [29]. ICA components meeting these criteria were automatically selected for the following analysis.

*Peak detection*. We performed peak detection on the selected ICA components time series. For peak detection, data were filtered in the 20–90 Hz spectral band. Additionally, we preprocessed ICA components' time series using sklearn RobustScaler to apply amplitude-based peak detection with the same parameters for different patients. Peak detection was performed using the peak-finding routine *scipy.signal.find_peaks()* implemented in SciPy. To avoid different settings for different patients, we automatically decreased the threshold until at least 300 peaks were detected in each patient.

*Peak selection*. Detected peaks underwent dipole fitting by MUSIC algorithm [29] (Mosher and Leahy 1999), computed on -20 to 30 ms around the spikes peak. To avoid redundancy in the data, only the event with the largest GOF was selected in each 0.5 s time window. In other words, the interspike interval was forced to be larger than 0.5 s. The output of stage I was a set of timestamps of identified spikes candidates.

**2.6.2 Stage II.** *αCSC decomposition*. We applied an optimization framework of convolutional sparse coding known as αCSC. Specifically, we used a multivariate model with rank-1 constraint [19], which reflects the fact that one source can be observed on a manifold of MEG sensors. In a nutshell, the multivariate time series data $X^n$ [$n \times T$], with $n$ number of sensors and $T$ the total time of the recording, is decomposed in a series of $k$ atoms with spatial pattern $u_k$ [$n \times k$] and temporal pattern $v_k$ [$k \times t$], with $t$ the duration of one event. The proximity of each data point to the k-th atom is defined by the activation vector $z_k$ [$k \times T$], which is the element introducing sparsity. $z_k$ has only few non-zero entries and it is always positive, meaning

that one atom repeats along the time series with the same polarity. The temporal extension of each atom $v_k$ was set to t = 0.5 s, and the regularization parameter ($\lambda$) set equal to 0.1. The mathematical formulation is summarized as follows

$$min_{u_k, v_k, z_k^n} \sum_{n=1}^{N} \frac{1}{2} \|X^n - \sum_{k=1}^{K} z_k^n * (u_k v_k^\top)\|_2^2 + \lambda \sum_{k=1}^{K} \|z_k^n\|_1,$$

$$s.t. \|u_k\|_2^2 \leq 1, \|v_k\|_2^2 \leq 1 \ and \ z_k^n \geq 0.$$

Before applying convolutional sparse coding, the data was bandpassed in the 2–90 Hz spectral range.

*Spatio-temporal clustering.* It was recently shown that αCSC can automatically detect biological artifacts and non-sinusoidal patterns [19,20] (Jas et al. 2017; La Tour et al. 2018). In our pipeline, the number of atoms in the dictionary was restricted to 3 atoms for each sensor type. To maximize the performance of αCSC, we concatenated epochs of 1s centered on the time-stamps provided from stage I. This allowed us to use αCSC purely as a clustering technique, where each cluster is identified by one atom of the library.

*Events selection.* The αCSC algorithm returns spatio-temporal patterns as atoms, and provides, for each atom $k$, a spatial pattern $u_k$ (weights of each sensor), a temporal pattern $v_k$ (temporal time trend) and an activation vector $z_k$, which defines the proximity of each time point of the MEG time series to the atom. To assign events to each atom, we thresholded at 7 median absolute deviations (MAD), and iteratively decreased the threshold until either at least 15 events were selected or the threshold reached 1.5 MAD of $z_k$. The derived epochs were assigned to the atom k. Each atom was therefore linked to a set of events with similar spatio-temporal patterns. This stage provides spatially aligned and clustered events.

*Atoms library generation.* We iterated our pipeline 4 times through stages 1 and 2. The rationale for multiple iteration is that prominent ICA components might miss relevant information. Therefore, we include additional runs forcing the clustering of spike events around different subsets of ICA components. In the first run, we include all ICA components passing the Kurtosis and GOF criteria, and extract the first three atoms. In the following three runs, we include a subset of ICA components showing similar topography (k-means based clustering). For each run, we extract three atoms. We constrained αCSC to fit three atoms in order to extract stable patterns in a reasonable time (20 minutes, 204 chs, sampling frequency = 200 Hz, 3 atoms: 5 mins). Therefore, after 4 runs, we obtain twelve candidate atoms. The goodness of the cluster built around each atom was estimated according to three features: GOF of the spatial pattern $u_k$; mean temporal correlation of the $v_k$ with the epoch time series from the sensor with maximum $u_k$; number of events in the cluster (= 1 for 20 or more). Each of these three features was scored from 0 to 1 and their average represented the atom's score. The atoms exceeding the mean+1standard deviation of the atoms score distribution were selected to populate the atoms' library. The selected atoms were visually reviewed, and atoms with unclear patterns were excluded (17/42 atoms were excluded in the analyzed dataset).

**2.6.3 Source reconstruction.** To delineate the irritative zone we used Minimum Norm Estimation (MNE, implemented in MNE-Python [26]. The covariance matrix was computed in the interval [-0.5 0.5] seconds around the spike peak of the averaged events in each cluster. The noise covariance was considered diagonal.

Visually marked spikes were individually reconstructed at peak latency [30]. The activation map for each spike was binarized, with value 1 for each vertex exceeding 50% of the maximum activation and 0 otherwise. The final binary map contained only locations pointed by at least half of the individual spike maps. Finally the binary map was smoothed within 10 mm in order to delineate the region associated with the spiking activity.

Spatiotemporal clusters based on αCSC atoms were used to estimate the irritative zone. For each atom in the library, the average of all clustered events was localized [31]. Atoms resulting from gradiometers and magnetometers were treated as independent contributions. An activation map was computed for each atom at two latencies identified as PEAK, i.e. the latency of maximum amplitude of the sharp deflection of the spike, and SLOPE, identified as the latency preceding the PEAK where the activity is still above baseline and the spatial pattern provides a distinct focus (see also supplementary material, S2 Fig). In both cases, SLOPE and PEAK, the activation map was thresholded at 50% of the maximum activation value, and translated into a binary activation map (1 if activation is higher, 0 otherwise, as in [30]). For each patient, we summed the binary maps from all atoms. Sources pointed by more than half of the atoms were selected and smoothed in the range of 10 mm. The resulting map delineated the predicted irritative zone.

## 2.7 Statistics

For each patient we obtained four anatomical maps: the resected area (RA), the irritative zones delineated by visually marked spikes, the irritative zone identified by automatically identified spikes at the level of the ascending slope (SLOPE) and at the level of the peak (PEAK).

We validate these three approaches by computing the average distance of the estimated irritative zone to the border of the resected area [32]. The RA was converted into a convex hull and the average distance of each point of the estimated irritative zone from the hull surface was computed. So the more negative the distance, the more is the overlap of the estimated irritative zone with the resected area; the more positive the distance, the larger the discrepancy between the estimated irritative zone and the resected area.

The pipeline is implemented as a python package megspikes which can be found at https://github.com/MEG-SPIKES/megspikes. The code for reproducing the analysis and figures can be found at https://github.com/MEG-SPIKES/aspire-alphacsc-epilepsy-MEG.

## 3 Results

We automatically detected spike clusters in 20 minutes of sleep from seven patients with good surgery outcome (Table 1), following our two-steps procedure: identification of candidate epileptiform events and events clustering by convolutional sparse coding. We indicate here with atom the spatiotemporal pattern defining a cluster of epileptic spikes. In Fig 2 we show the example of one atom and the events populating the atom's cluster: spatial (Fig 2A) and temporal (Fig 2B) patterns characterize the profile of repeating epileptic events. The average of the events displayed on the MEG layout shows a focal activation over the left hemisphere (Fig 2C). Single events presented a stable temporal profile in the sensors with largest activation (Fig 2D). In the seven cases analyzed, we identified 25 atoms (16 from gradiometers, 9 from magnetometers), with an average of 3.6 [range 1–6] atoms per case (2.3 [1.0–3.0] for gradiometers, 1.3 [1.2–3.0] for magnetometers). We detected 549 total spikes, reflecting a spike rate of 2.41 [0.9–3.6] spikes/minute (recording duration: 19.5 [12–24] minutes). We assigned an average of 22.1 [15.0–31.0] events to each cluster, with similar amounts of spikes detected from gradiometers (21.1 [15.0–30.0] spikes/cluster) and magnetometers (23.7 [17.0–31.0] spikes/cluster).

We validated the output of our analysis against the irritative zone identified by visual analysis, the resected area and the surgery outcome. In Fig 3 we illustrate the case of patient 5: resected area and visually marked spike locations are superimposed on the individual patient MR in Fig 3A, while the resected area and irritative zone identified by visual analysis projected on the modeled cortical surface are depicted in panel 3B and 3C respectively. The averaged time trend for one cluster is presented in the sensor space (panel 3D) and in the source space

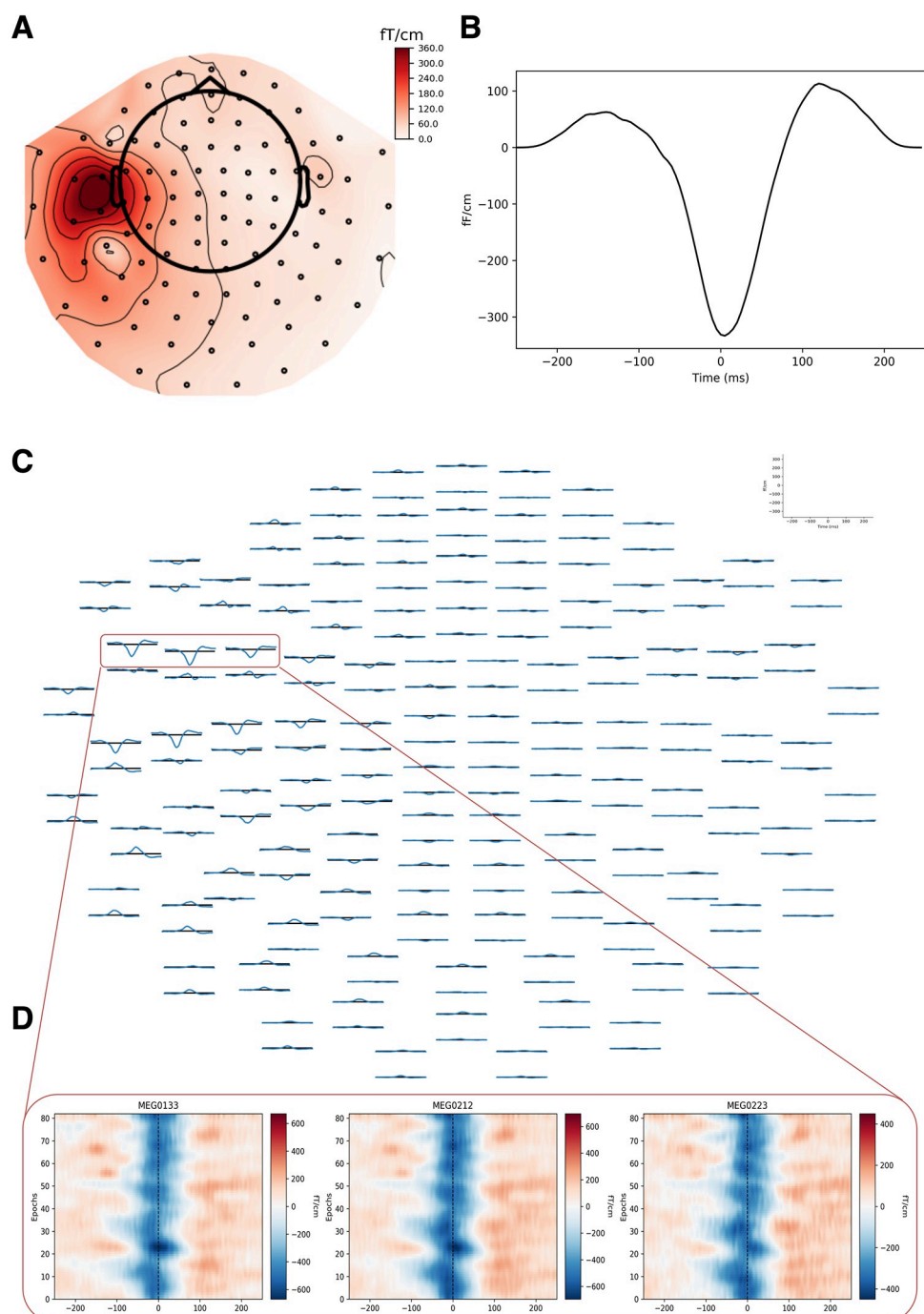

**Fig 2. Example of one atom from gradiometers in patient 7. A**. Atom's spatial pattern. **B**. Atom's temporal pattern. **C**. Average of the epileptiform events assigned to this cluster. **D**. Temporal pattern of events from three representative gradiometers (x-axis:time, y-axis:event number).

(panel 3E), while the cortical spread of activation at SLOPE and PEAK latencies (see methods 2.6.3) is presented in 3F. Those activations anatomically define the predicted irritative zone, which is the output of our analysis pipeline.

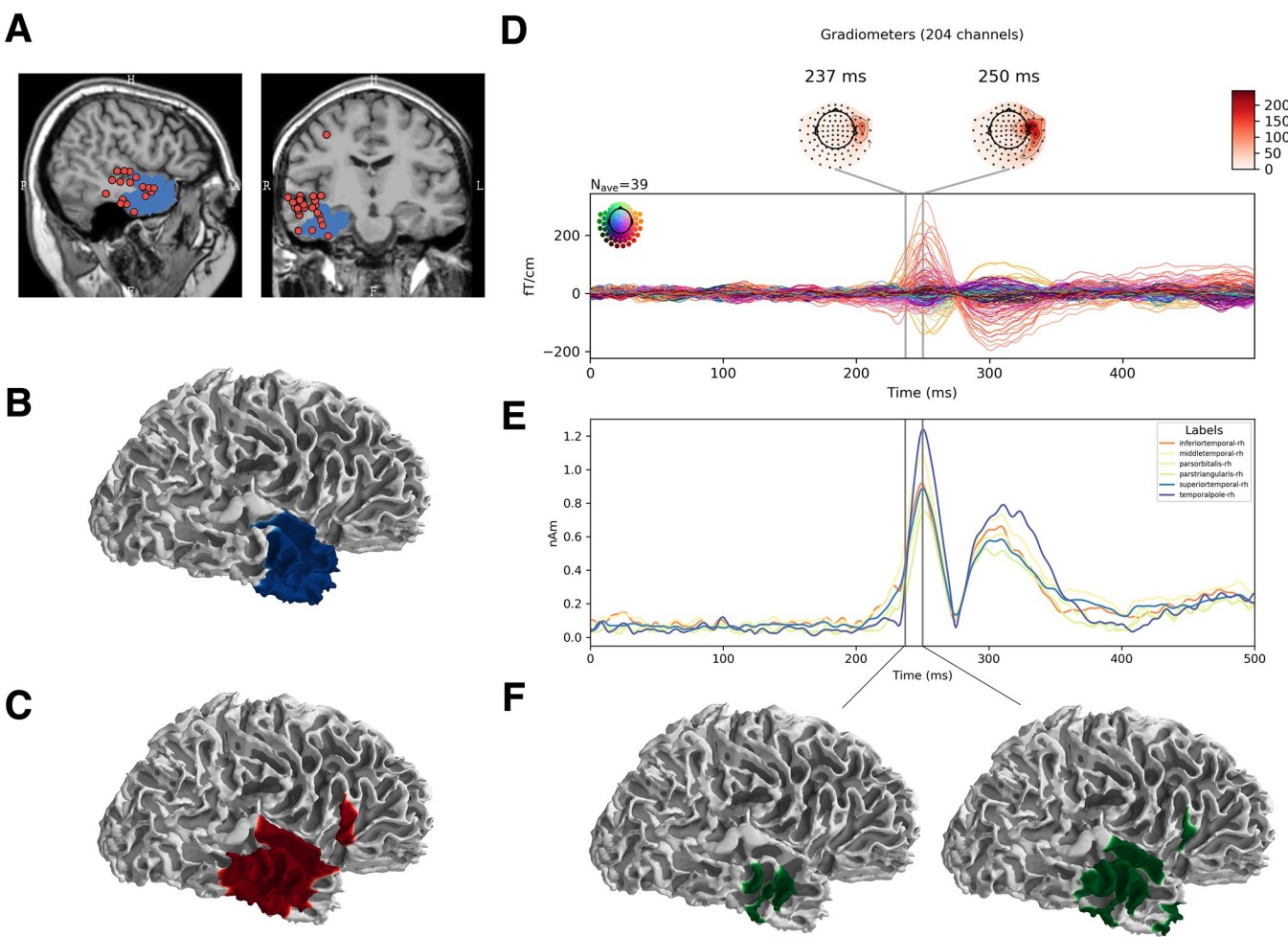

**Fig 3. Illustration of the performance of our automated epileptic spike pattern detector applied on pre-surgical MEG data from patient 5. A**. Localization of visually detected spikes with single equivalent dipole fit (red dots) in the Elekta software and the resection area (blue area); **B**. Surgically resected area (lateral view, right hemisphere); **C**. Irritative zone defined by Minimum Norm Estimation (MNE) on visually detected spikes; **F (left)**. Irritative zone defined by (MNE) along the ascending slope of the averaged automatically detected spike cluster; **F (right).** Irritative zone defined by (MNE) on the peak of the averaged automatically detected spike cluster; **D**. Sensor space butterfly plot of the averaged automatically detected spike cluster; **E**. Source space time series of the automatically detected spike cluster. Traces correspond to 6 anatomical locations (FreeSurfer segmentation) with the highest averaged activity in the source space.

In this case, the automated procedure identified the source of epileptic activity within the resected area. The patient had a good surgery outcome (see Table 1), which exemplifies the clinical relevance of our approach.

We identified cortical sources of cluster averages at the amplitude SLOPE and PEAK for all patients and therefore delineated our predicted irritative zone. To quantify the reliability of our predicted irritative zone, first we visually confirmed the sublobar concordance with the resected area, as in [32]. We observed an overlap of the resected area with both visually identified and automatically detected irritative zones in six out of seven patients. In one patient (patient 7), we observed concordance between visually identified and automatically detected irritative zones, both not overlapping with the resected area, and the patient had outcome Engel IIB.

SLOPE and PEAK latencies of the detected sources correspond to different spatial spread of activity, and therefore to different localization and extent of the irritative zone (Fig 4). We

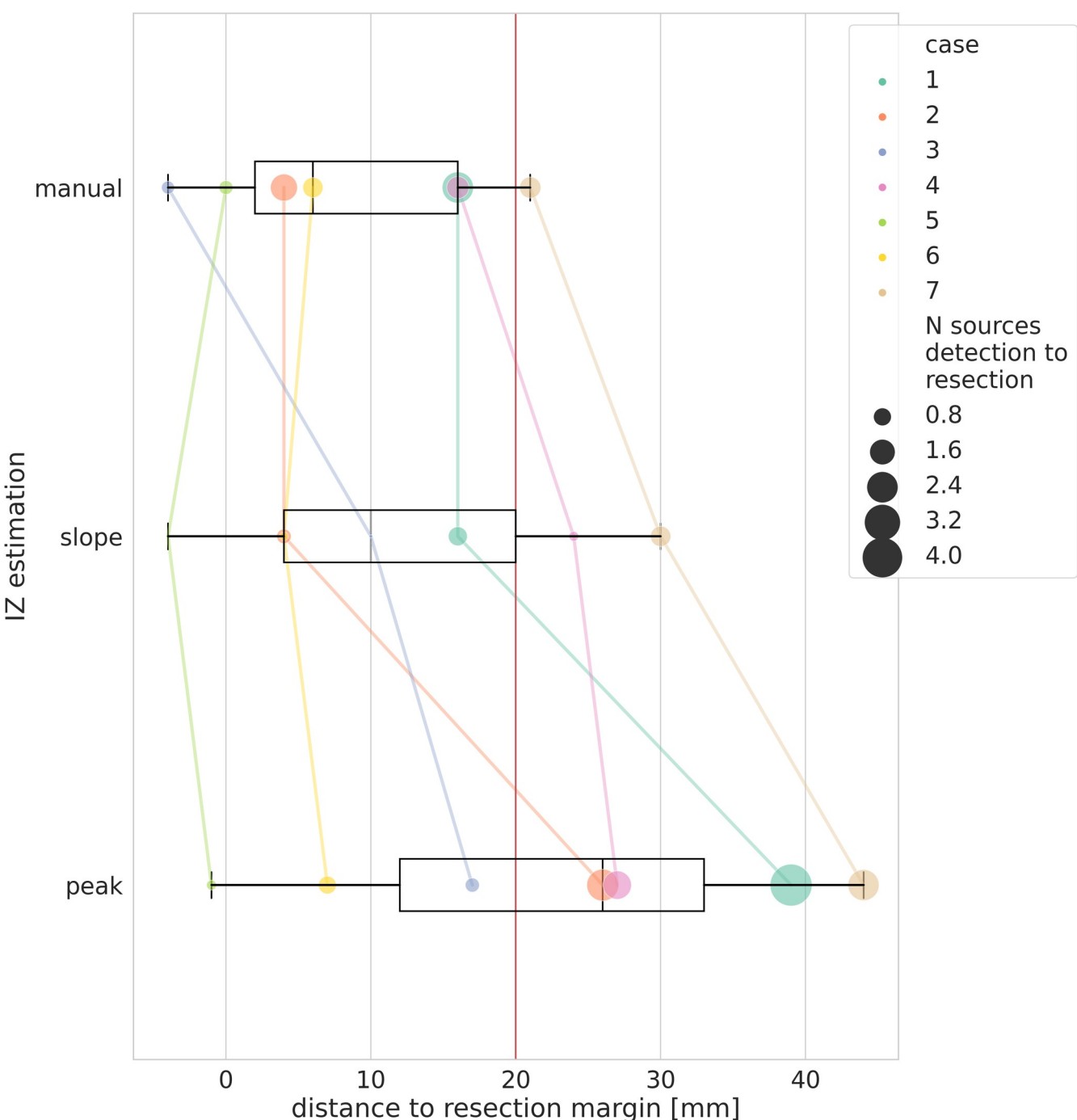

**Fig 4. Accuracy of different estimations of the IZ.** The average distance of the estimated IZ from the resection border is represented on the x-axis. The IZ estimated by the automated clustering at the level of the ascending slope, the visually marked events, and the automated clustering at the level of the peak, are shown for each case. Each patient is represented with a color and the size of the dot is proportional to the ratio between the size of the estimated IZ and the size of the resected area, to provide a comparison within and between patients. The red line corresponds to the 20 mm average distance from the resection border.

compared the average distance of the irritative zone from the resection border at the SLOPE and PEAK latencies and observed that that irritative zone detected at the SLOPE is significantly closer to the resection border (Wilcoxon signed rank test, p = 0.01). SLOPE based

distance from the resection border was not significantly distinguishable from the distance based on visual analysis (Wilcoxon signed rank test, p = 0.31). Across patients, average distance from the resection border was 8.4 ± 9.3 mm for visual analysis, 12.0 ± 12.0 mm for SLOPE and 22.7 mm ± 16.4 mm for PEAK. Therefore, the distance of the IZ at both SLOPE and PEAK latencies is relatively close to the resection margin. However, only when considering the source activation at the spike onset, the performance of the automated detection is similar to the expert observer.

## 4 Discussion

We proposed here a data driven approach for the automated identification of epileptiform activity patterns in MEG data. We showed that selection and clustering of events based on dictionary learning correctly supported the delineation of the irritative zone in the single patient for cases with good surgical outcome. Our approach provided an estimation of the irritative zone consistent with the visual inspection of expert observers. Therefore, our results support the standardization of the analysis of interictal epileptiform activity in MEG data.

We followed a two-step procedure to select the events of interest and then cluster them. Our approach follows a procedure similar to spike sorting in microelectrodes recordings [33], where detected neuronal spike peaks are used to build spatio-temporal templates by density-based clustering. Clustering epilepeptic spikes in MEG recording represents a more challenging problem, given the lower number of events and their signal to noise ratio (SNR). On the other hand, given the biophysical relation between source and sensor space in MEG data, we optimized this scheme for the delineation of the irritative zone in non-invasive human epilepsy data. The data feed into the pipeline were visually screened to minimize the presence of artifact. The peak detection was performed on ICA components mimicking sources with high kurtosis. Searching for peaks in source space rather than in sensor space signal improves signal-to-noise ratio (SNR) and is more robust to artifacts. As an advancement on [21] (Ossadtchi et al. 2004), we did not cluster separately with respect to spatial and temporal source features of each single event, but we rather exploited the consistency of the spatiotemporal pattern of multiple events, which led to the estimation of data-driven 'atoms'. Convolutional sparse coding is based on alpha stable heavy tailed noise distributions, which makes it robust against bursty events. The 'atoms' defined the library of learned repeating events, with each 'atom' representing the center of each cluster in feature space.

Previously proposed detector strategies relied on statistical properties in the time, frequency and time-frequency domain [34], either on single channel or multivariate data [12]. Recent approaches implemented machine learning to automatically classify epileptic events. Deep learning networks recognizing time domain morphological features [35] and based on short-term memory networks [36], as well as convolutional neural network processing spectral information [37] have been proven as valuable approaches on EEG data. Deep learning has been successfully applied also on a MEG dataset of 20 patients affected by focal epilepsy [15]. Compared to other proposed approaches, dictionary learning does not require specific assumptions on the spike statistics and is not dependent on algorithm training on an extensive dataset. Therefore, it can be prospectively applied to the next incoming case.

The opportunity to standardize interictal events detection might further increase the added value of MEG recordings in epilepsy investigation. To date, spikes and sharp waves identify the irritative zone, which can overlap with the seizure onset zone and support the delineation of the resection margins [38] (Jehi 2018). The clinical value of the irritative zone has been validated on intracranial EEG [7,8] where seizure freedom was associated with the resection of dipole sources identified in MEG data. In prospective studies, MEG analysis outcome

contributed to improve the subsequent implantation [9,10]. In a meta analysis of 6 clinical studies, it was shown that MEG source localization of interictal events might be even more specific that stereoEEG [39]. While the use of MEG improves clinical practice, neurologists are involved in time-consuming bias-prone data inspection. Ideally, a clinical MEG recording session should lead to the identification and localization of a sufficient number of IED sources and to delineation of a reliable and reproducible spatiotemporal map of the irritative zone. In this context, an automated approach based on peak analysis might overestimate the extension of the irritative zone. Clustering events allows access to source activity at latencies preceding the spike peak with higher SNR. The proposed approach quickly provides neurologists with the activation profile of candidate clusters, which can be inspected and possibly contribute to clarify the structure of the epileptic network.

The identification of epileptiform activity is a crucial step to anatomically target spike generators. Source estimation of interictal pattern in MEG has been validated by comparison with intracranial recordings [30,40,41]. In this project we used distributed sources imaging, which was shown to be more informative than equivalent current dipole [32]. However, while source estimation is reliable for stationary activity, spike propagation represents still one major concern for the delineation of the epileptic network and the identification of the seizure onset zone. Therefore, rather than the spatial extent of the irritative zone, its intrinsic dynamic must be characterized. To date, the resection of sources of propagating spike patterns identified in invasive stereoEEG recordings are associated with good surgery outcome [42,43]https://www.zotero.org/google-docs/?T7WkMp. The clinical analysis of MEG interictal patterns should therefore localize spike generators at latencies preceding the spike peak, where the SNR is higher, in order to account for spatial propagation. The consideration of the ascending slope in MEG has proven strong association of the irritative zone with good outcome [11], while recent evidence from high density EEG and MEG identifies the highest accuracy in the earliest resolvable phase of the IED onset [44]. Source estimation of the single event for latencies preceding the spike peak might be affected by poor SNR [45,46]. Therefore, the opportunity to cluster similar events and consider their spatio-temporal averaged pattern empowers our ability to localize early activations. Our approach allows us to consider the evolution of the activity spread along the rising slope of the averaged time trend for each cluster of the irritative zone. Enhancing the SNR of repeating patterns, we might provide a finer localization of the event onset.

## Limitations

We demonstrated the applicability of convolutional sparse coding for spike detection and localization of the irritative zone in epileptic patients. However, the limited dataset considered represents the framework only for a technical validation. While we could reproduce the findings of visual analysis and provide clinically relevant information, a larger set of cases is needed to further quantify the reliability of our approach and verify its implementation in the clinical settings.

## Conclusions

We propose here the validation of an algorithmic approach for the detection and clustering of IEDs, emulating the analysis of expert reviewers consistently with the clinical findings. The opportunity to model epileptiform activity into spatiotemporal 'atoms' enhances SNR and provides reliable source localization at the early stage of the IED onset. This, in turn, might support clinicians in the delineation of the irritative zone, providing automated access to spatial and temporal features characterizing the epileptic network.

## Supporting information

**S1 Fig. Selection of "Peaky" ICA components.** In case #4, the ICA decomposition of magnetometers provided components 3 and 4, both featuring a dipolar spatial pattern (GOF ICA3 = .96, GOF ICA4 = 0.98). However, only ICA component 2 presented spike-like events in the temporal pattern, which was reflected in a Kurtosis value of 5.6, while the Kurtosis for ICA component 3 as 0.7. The code to reproduce this image is published at https://github.com/MEG-SPIKES/aspire-alphacsc-epilepsy-MEG/blob/main/analysis/05_revision.ipynb.
(TIFF)

**S2 Fig. Distance from the resection margin from take-off to peak latency of the averaged clusters.** Slope 1–5 refers to latencies between the take-off and the 50% slope. Slope 6–9 refers to latencies between the 50% slope and the peak. The latencies were individually selected for each cluster because of the different length of the ascending slopes. The code to reproduce this image is published at https://github.com/MEG-SPIKES/aspire-alphacsc-epilepsy-MEG/blob/main/analysis/05_revision.ipynb.
(TIFF)

**S3 Fig. Resected volume and irritative zone of each patient computed according to manual spike marking, automated peak and slope estimation.** Pages 1–7 correspond to patient 1–7 in Table 1.
(PDF)

## Author Contributions

**Conceptualization:** Valerii Chirkov, Tatiana Stroganova, Alexei Ossadtchi, Tommaso Fedele.

**Data curation:** Valerii Chirkov, Anna Kryuchkova, Alexandra Koptelova, Tatiana Stroganova, Tommaso Fedele.

**Formal analysis:** Valerii Chirkov, Anna Kryuchkova, Tatiana Stroganova, Alexandra Kuznetsova, Daria Kleeva, Tommaso Fedele.

**Funding acquisition:** Tatiana Stroganova, Alexei Ossadtchi, Tommaso Fedele.

**Methodology:** Valerii Chirkov, Anna Kryuchkova, Tatiana Stroganova, Alexandra Kuznetsova, Alexei Ossadtchi, Tommaso Fedele.

**Resources:** Tatiana Stroganova.

**Software:** Valerii Chirkov, Alexei Ossadtchi, Tommaso Fedele.

**Supervision:** Tatiana Stroganova, Alexei Ossadtchi, Tommaso Fedele.

**Writing – original draft:** Valerii Chirkov, Tommaso Fedele.

**Writing – review & editing:** Tatiana Stroganova, Alexei Ossadtchi, Tommaso Fedele.

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
