## [Decision Letter · Decision Letter 0]

31 Mar 2022

PONE-D-22-01045Data-driven approach for the delineation of the irritative zone in epilepsy in MEGPLOS ONE

Dear Dr. Fedele,

Thank you for submitting your manuscript to PLOS ONE. After careful consideration, we feel that it has merit but does not fully meet PLOS ONE’s publication criteria as it currently stands. Therefore, we invite you to submit a revised version of the manuscript that addresses the points raised during the review process.

The manuscript should be significantly revised to address the comments provided by the reviewers.

In the discussion section, the authors should justify scientific value or practical advantages of their approach.Based on the study from Jas et al. (2017), the spatiotemporal clustering method used in this study (convolutional sparse coding, CSC) could be sensitive to artefacts that are common in MEG recordings of active epilepsy patients. How are those artefacts handled in this study?After clustering IED events in magnetometers and gradiometers separately, were the same events combined/aligned? If so, how were the same events identified separately aligned based on peak time point? How did this affect the MNE final solution?In Page 6 section 2.6 line 3, Yger et al. 2018 (typo as well i.g should be i.e.), how does the Author's approach actually differ from the Yger approach that is referred to as being "conceptually related"? Please clarify this point.In Page 6 section 2.6.1 components selection, do authors quantify the variance of components selection between raters, as the selection process could be subjective and bias may be introduced?The spatiotemporal clustering method used in this study (convolutional sparse coding, CSC, La Tour et al., 2018) has been accepted by NeurIPS 2018. Please cite the conference paper, instead of BioRxiv preprint.Can the Authors be less subjective in their description of 'peakiness' for their ICA characterisation. It is not clear to the reader how this looks when differentiating between less peaky and more peaky components. An image might help.With means at VIS 8.4mm, SLOPE 19.7mm, PEAK 30.7mm from the resection margins, the Authors conclude that "consideration of source spread at the ascending slope provided an IZ closer to RA (resection area) and resembled the analysis of the expert observer". I think the term 'resemble' is an overstatement as there is a clear gap between the VIS and the automated results.Further to Point 7, to argue that results would be closer to the accuracy of the VIS results closer to the take-off is conjectural. Why not run a further analysis closer to take-off (eg. half-way between take-off and the 50% rising phase?And further to Points 7 and 8, the logic of 'source spread' as an explanation for the differences between VIS and Automated appears to be contradicted by the fact that VIS results appeared to be based on spike peaks (Section 2.6.3, line 5). This needs to be addressed in the Discussion.Only < 2 mins recording for the automated analysis. How long were the full recordings? How can the Authors be certain that this very short period was representative of the entire recording?We recommend to remove the word 'innovative' from the text as this is an opinionated term.

We look forward to receiving your revised manuscript.

Kind regards,

Gennady S. Cymbalyuk, Ph.D.

Academic Editor

PLOS ONE

Journal Requirements:

2. Please provide additional details regarding participant consent. In the Methods section, please ensure that you have specified (1) whether consent was informed and (2) what type you obtained (for instance, written or verbal). If your study included minors, state whether you obtained consent from parents or guardians. If the need for consent was waived by the ethics committee, please include this information.

“V.K., A.Kr., and T.F. were supported by Russian Foundation for Basic Research, N◦20-015-00176 A (https://kias.rfbr.ru/). The funders had no role in study design, data collection and analysis, decision to publish, or preparation of the manuscript.“

“V.K., A.G., and T.F. were supported by Russian Foundation for Basic Research, N◦20-015-00176 A (https://kias.rfbr.ru/).  The funders had no role in study design, data collection and analysis, decision to publish, or preparation of the manuscript.”

Reviewers' comments:

Reviewer's Responses to Questions

**Comments to the Author**

1. Is the manuscript technically sound, and do the data support the conclusions?

Reviewer #1: Yes

Reviewer #2: Partly

2. Has the statistical analysis been performed appropriately and rigorously? 

Reviewer #1: Yes

Reviewer #2: Yes

3. Have the authors made all data underlying the findings in their manuscript fully available?

Reviewer #1: No

Reviewer #2: Yes

4. Is the manuscript presented in an intelligible fashion and written in standard English?

Reviewer #1: Yes

Reviewer #2: Yes

5. Review Comments to the Author

Reviewer #1: The work is about delineating from sensor-level MEG data the brain region in epilepsy, commonly called irritative zone that is at or near the epileptic seizure origin. The authors used ICA analyses, selected ICA components by convolutional sparse coding and computed the distances to the spatial map or maps of the peaky ICA components from the resected regions of the patients retrospectively. The localization results are not better than visually identified locations by two to three folds.

I do not see any scientific value of these findings, or the novelty of the method, and do not find these findings important enough for a scientific publication.

Reviewer #2: The Authors have retrospectively studied MEG-recorded IEDs in 7 epilepsy surgery patients to try to reliably identify the irritative zone (IZ) based on proximity of MNE results to the resection linked to good outcomes. They did this by visually marking IEDs (VIS), and compared this with the marking of ICA informed source space signals at the halfway point (SLOPE) and then at the PEAK (using distance from resection as the clinical reference). These signals were first identified by their ICA 'peakiness' and then clustered according to their spatiotemporal pattern using 'convoluted sparse coding'. An average of 3.6 of these ST pattern clusters per patient were found (averaging 22 spikes per cluster). It seems that the automated assessment did not perform as well as the visual based analysis (averages VIS 8.4mm, SLOPE 19.7mm, PEAK 30.7mm from the resection margins).

They conclude that "consideration of source spread at the ascending slope provided an IZ closer to RA (resection area) and resembled the analysis of the expert observer".

I think this paper does address an important logistical problem in the clinical use of these large MEG datasets for patients undergoing epilepsy surgery work up. While expert assessment will always be important, more efficient ways of analysis are needed. At this point, there is no agreed upon method of automated analysis of these datasets.

While the Authors do seem to describe and execute a sound approach to the problem from the technical perspective, I think there are some points I raise below in an effort to perhaps improve the transparency and the quality of the work.

1. Based on the study from Jas et al. (2017), the spatiotemporal clustering method used in this study (convolutional sparse coding, CSC) could be sensitive to artefacts that are common in MEG recordings of active epilepsy patients. How are those artefacts handled in this study?

2. After clustering IED events in magnetometers and gradiometers separately, were the same events combined/aligned? If so, how were the same events identified separately aligned based on peak time point? How did this affect the MNE final solution?

3. In Page 6 section 2.6 line 3, Yger et al. 2018 (typo as well i.g should be i.e.), how does the Author's approach actually differ from the Yger approach that is referred to as being "conceptually related"? Please clarify this point.

4. In Page 6 section 2.6.1 components selection, do authors quantify the variance of components selection between raters, as the selection process could be subjective and bias may be introduced?

5. The spatiotemporal clustering method used in this study (convolutional sparse coding, CSC, La Tour et al., 2018) has been accepted by NeurIPS 2018. Please cite the conference paper, instead of BioRxiv preprint.

6. Can the Authors be less subjective in their description of 'peakiness' for their ICA characterisation. It is not clear to the reader how this looks when differentiating between less peaky and more peaky components. An image might help.

7. With means at VIS 8.4mm, SLOPE 19.7mm, PEAK 30.7mm from the resection margins, the Authors conclude that "consideration of source spread at the ascending slope provided an IZ closer to RA (resection area) and resembled the analysis of the expert observer". I think the term 'resemble' is an overstatement as there is a clear gap between the VIS and the automated results.

8. Further to Point 7, to argue that results would be closer to the accuracy of the VIS results closer to the take-off is conjectural. Why not run a further analysis closer to take-off (eg. half-way between take-off and the 50% rising phase?

9. And further to Points 7 and 8, the logic of 'source spread' as an explanation for the differences between VIS and Automated appears to be contradicted by the fact that VIS results appeared to be based on spike peaks (Section 2.6.3, line 5). This needs to be addressed in the Discussion.

10. Only < 2 mins recording for the automated analysis. How long were the full recordings? How can the Authors be certain that this very short period was representative of the entire recording?

11. I would remove the word 'innovative' from the text as this is an opinionated term.

6. PLOS authors have the option to publish the peer review history of their article (what does this mean?). If published, this will include your full peer review and any attached files.

Reviewer #1: No

Reviewer #2: **Yes: **(Professor) Chris Plummer

---

## [Author Response · Author response to Decision Letter 0]

19 May 2022

We thank the reviewers for their valuable comments.

We answer here point by point to the reviewers comments and provide the link to changes in the manuscript in the file "Response to Reviewers"

---

## [Decision Letter · Decision Letter 1]

19 Jul 2022

PONE-D-22-01045R1Data-driven approach for the delineation of the irritative zone in epilepsy in MEGPLOS ONE

Dear Dr. Fedele,

Thank you for submitting your manuscript to PLOS ONE. After careful consideration, we feel that it has merit but does not fully meet PLOS ONE’s publication criteria as it currently stands. Therefore, we invite you to submit a revised version of the manuscript that addresses the points raised during the review process.

Please, adequately address the concern of misinterpreting the results of the paper by Plummer et al.  

We look forward to receiving your revised manuscript.

Kind regards,

Gennady S. Cymbalyuk, Ph.D.

Academic Editor

PLOS ONE

Journal Requirements:

Reviewers' comments:

Reviewer's Responses to Questions

**Comments to the Author**

1. If the authors have adequately addressed your comments raised in a previous round of review and you feel that this manuscript is now acceptable for publication, you may indicate that here to bypass the “Comments to the Author” section, enter your conflict of interest statement in the “Confidential to Editor” section, and submit your "Accept" recommendation.

Reviewer #1: All comments have been addressed

Reviewer #2: (No Response)

2. Is the manuscript technically sound, and do the data support the conclusions?

Reviewer #1: Yes

Reviewer #2: Yes

3. Has the statistical analysis been performed appropriately and rigorously? 

Reviewer #1: Yes

Reviewer #2: Yes

4. Have the authors made all data underlying the findings in their manuscript fully available?

Reviewer #1: No

Reviewer #2: Yes

5. Is the manuscript presented in an intelligible fashion and written in standard English?

Reviewer #1: Yes

Reviewer #2: Yes

6. Review Comments to the Author

Reviewer #1: In this study, the authors (i) detected and analyzed 20 mins of MEG recordings of the interictal activity from seven patients affected by epilepsy who had undergone successful epilepsy surgeries and (ii) mapped the seizure onset zone or irritative zone with various measures approximately within a cm to four cm distance from the resected regions.

Reviewer #2: In relation to this response:

"Further to Point 7, to argue that results would be closer to the accuracy of the VIS

results closer to the take-off is conjectural. Why not run a further analysis closer to take-off (eg.

half-way between take-off and the 50% rising phase?

RESPONSE: We share the reviewer’s interest in the information contained in the takeoff

latency and we tested the possibility to extract valuable information from earlier

latencies of the ascending slope. We illustrate the results of this analysis in the figure,

where we show stable SNR for latencies selected at 50% slope and later, while between

take off and 50% slope the reconstruction does not provide valuable clinical information.

We find our results in line with Plummer et al., 2019.

(I have read over this paper (by Plummer et al) which actually does find valuable clinical information between take off and the 50% slope (so not in line with the paper by Plummer et al in this respect)

We provide the outcome of our analysis in Figure S2 in Supplementary Material, and

refer to it in the Methods, section 2.6.3:

“An activation map was computed for each atom at two latencies identified as PEAK, i.e.

the latency of maximum amplitude of the sharp deflection of the spike, and SLOPE,

identified as the latency preceding the PEAK where the activity is still above baseline

and the spatial pattern provides a distinct focus (see also supplementary material, figure

S2)”

Plummer, Chris, Simon J. Vogrin, William P. Woods, Michael A. Murphy, Mark J. Cook, and

David T.J. Liley. 2019. “Interictal and Ictal Source Localization for Epilepsy Surgery Using High-

Density EEG with MEG: A Prospective Long-Term Study.” Brain 142 (4): 932–51.

https://doi.org/10.1093/brain/awz015.

7. PLOS authors have the option to publish the peer review history of their article (what does this mean?). If published, this will include your full peer review and any attached files.

Reviewer #1: No

Reviewer #2: **Yes: **Chris Plummer

---

## [Author Response · Author response to Decision Letter 1]

12 Aug 2022

We thank the reviewers for their valuable comments.

As requested, we address the concern of misinterpreting the results of the paper by Plummer et al.

Reviewer comment (all text below is from the Editor and Reviewers letter).

“””

Reviewer #2: In relation to this response:

"Further to Point 7, to argue that results would be closer to the accuracy of the VIS

results closer to the take-off is conjectural. Why not run a further analysis closer to take-off (eg.

half-way between take-off and the 50% rising phase?

RESPONSE: We share the reviewer’s interest in the information contained in the takeoff

latency and we tested the possibility to extract valuable information from earlier

latencies of the ascending slope. We illustrate the results of this analysis in the figure,

where we show stable SNR for latencies selected at 50% slope and later, while between

take off and 50% slope the reconstruction does not provide valuable clinical information.

We find our results in line with Plummer et al., 2019.

(I have read over this paper (by Plummer et al) which actually does find valuable clinical information between take off and the 50% slope (so not in line with the paper by Plummer et al in this respect)

We provide the outcome of our analysis in Figure S2 in Supplementary Material, and

refer to it in the Methods, section 2.6.3:

“An activation map was computed for each atom at two latencies identified as PEAK, i.e.

the latency of maximum amplitude of the sharp deflection of the spike, and SLOPE,

identified as the latency preceding the PEAK where the activity is still above baseline

and the spatial pattern provides a distinct focus (see also supplementary material, figure

S2)”

Plummer, Chris, Simon J. Vogrin, William P. Woods, Michael A. Murphy, Mark J. Cook, and

David T.J. Liley. 2019. “Interictal and Ictal Source Localization for Epilepsy Surgery Using High-

Density EEG with MEG: A Prospective Long-Term Study.” Brain 142 (4): 932–51.

https://doi.org/10.1093/brain/awz015.

“””

OUR RESPONSE: We thank the reviewer for pointing to the difference with Plummer et al. 2019. We know acknowledge this difference in the Discussion session

The consideration of the ascending slope in MEG has proven strong association of the irritative zone with good outcome (Englot et al. 2015), while recent evidence from high density EEG and MEG identifies the highest accuracy in the earliest resolvable phase of the IED onset (Plummer et al. 2019).

While our focus in this project is the validation of spike detection and clustering, we plan to compare our approach with single event based analysis on a more comprehensive dataset which we are collecting for a follow-up study.

---

## [Decision Letter · Decision Letter 2]

12 Sep 2022

Data-driven approach for the delineation of the irritative zone in epilepsy in MEG

PONE-D-22-01045R2

Dear Dr. Fedele,

We’re pleased to inform you that your manuscript has been judged scientifically suitable for publication and will be formally accepted for publication once it meets all outstanding technical requirements.

Kind regards,

Gennady S. Cymbalyuk, Ph.D.

Academic Editor

PLOS ONE

Additional Editor Comments (optional):

Reviewers' comments:

Reviewer's Responses to Questions

**Comments to the Author**

1. If the authors have adequately addressed your comments raised in a previous round of review and you feel that this manuscript is now acceptable for publication, you may indicate that here to bypass the “Comments to the Author” section, enter your conflict of interest statement in the “Confidential to Editor” section, and submit your "Accept" recommendation.

Reviewer #2: All comments have been addressed

2. Is the manuscript technically sound, and do the data support the conclusions?

Reviewer #2: Yes

3. Has the statistical analysis been performed appropriately and rigorously? 

Reviewer #2: Yes

4. Have the authors made all data underlying the findings in their manuscript fully available?

Reviewer #2: Yes

5. Is the manuscript presented in an intelligible fashion and written in standard English?

Reviewer #2: Yes

6. Review Comments to the Author

Reviewer #2: I am happy that the Authors have addressed any outstanding comments. I think the paper addresses an important problem in this field of research.

7. PLOS authors have the option to publish the peer review history of their article (what does this mean?). If published, this will include your full peer review and any attached files.

Reviewer #2: No

---

## [Editor Report · Acceptance letter]

29 Sep 2022

PONE-D-22-01045R2 

Data-driven approach for the delineation of the irritative zone in epilepsy in MEG 

Dear Dr. Fedele:

I'm pleased to inform you that your manuscript has been deemed suitable for publication in PLOS ONE. Congratulations! Your manuscript is now with our production department. 

Kind regards, 

on behalf of

Dr. Gennady S. Cymbalyuk 

Academic Editor

PLOS ONE